# ToDD: Topological Compound Fingerprinting in Computer-Aided Drug Discovery

**Andac Demir**[*]
Novartis
andac.demir@novartis.com

**Baris Coskunuzer**[*]
University of Texas at Dallas
coskunuz@utdallas.edu

**Ignacio Segovia-Dominguez**
University of Texas at Dallas
Jet Propulsion Laboratory, Caltech

**Yuzhou Chen**
Temple University

**Yulia Gel**
University of Texas at Dallas
National Science Foundation

**Bulent Kiziltan**
Novartis
bulent.kiziltan@novartis.com

## Abstract

In computer-aided drug discovery (CADD), virtual screening (VS) is used for identifying the drug candidates that are most likely to bind to a molecular target in a large library of compounds. Most VS methods to date have focused on using canonical compound representations (e.g., SMILES strings, Morgan fingerprints) or generating alternative fingerprints of the compounds by training progressively more complex variational autoencoders (VAEs) and graph neural networks (GNNs). Although VAEs and GNNs led to significant improvements in VS performance, these methods suffer from reduced performance when scaling to large virtual compound datasets. The performance of these methods has shown only incremental improvements in the past few years. To address this problem, we developed a novel method using multiparameter persistence (MP) homology that produces topological fingerprints of the compounds as multidimensional vectors. Our primary contribution is framing the VS process as a new topology-based graph ranking problem by partitioning a compound into chemical substructures informed by the periodic properties of its atoms and extracting their persistent homology features at multiple resolution levels. We show that the margin loss fine-tuning of pretrained Triplet networks attains highly competitive results in differentiating between compounds in the embedding space and ranking their likelihood of becoming effective drug candidates. We further establish theoretical guarantees for the stability properties of our proposed MP signatures, and demonstrate that our models, enhanced by the MP signatures, outperform state-of-the-art methods on benchmark datasets by a wide and highly statistically significant margin (e.g., 93% gain for Cleves-Jain and 54% gain for DUD-E Diverse dataset).

## 1 Introduction

Drug discovery is the early phase of the pharmaceutical R&D pipeline where machine learning (ML) is making a paradigm-shifting impact [31, 91]. Traditionally, early phases of biomedical research involve the identification of targets for a disease of interest, followed by high-throughput screening

---

[*]Equal contribution.

36th Conference on Neural Information Processing Systems (NeurIPS 2022).

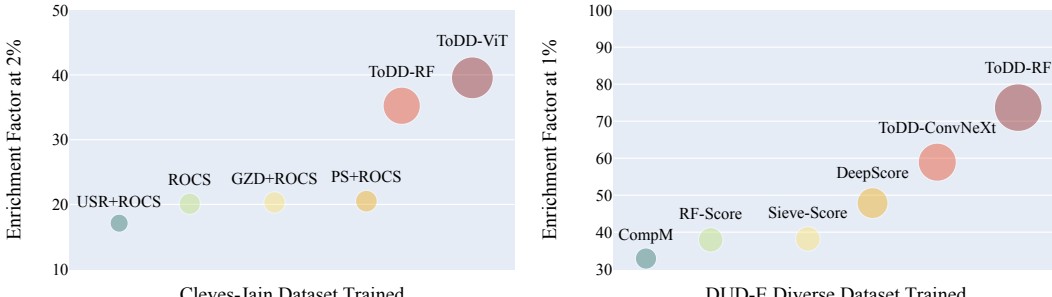

Figure 1: **Comparison of virtual screening performance.** Each bubble's diameter is proportional to its EF score. ToDD offers significant gain regardless of the choice of classification model such as random forests (RF), vision transformer (ViT) or a modernized ResNet architecture ConvNeXt. The standard performance metric $EF_{\alpha\%}$ is defined as $\frac{100}{\alpha}$, and therefore the maximum attainable value is 50 for $EF_{2\%}$, and 100 for $EF_{1\%}$.

(HTS) experiments to determine hits within the synthesized compound library, i.e., compounds with high potential. Then, these compounds are optimized to increase potency and other desired target properties. In the final phases of the R&D pipeline, drug candidates have to pass a series of rigorous controlled tests in clinical trials to be considered for regulatory approval. On average, this process takes 10-15 years end-to-end and costs in excess of $\sim 2$ billion US dollars [10]. HTS is highly time and cost-intensive. Therefore, it is critical to find good potential compounds effectively for the HTS step for novel compound discovery, but also to speed up the pipeline and make it more cost-effective. To address this need, ML augmented virtual screening (VS) has emerged as a powerful computational approach to screen ultra large libraries of compounds to find the ones with desired properties and prioritize them for experimentation [65, 40].

In this paper, we develop novel approaches for virtual screening by successfully integrating topological data analysis (TDA) methods with ML and deep learning (DL) tools. We first produce topological fingerprints of compounds as $2D$ or $3D$ vectors by using TDA tools, i.e., multidimensional persistent homology. Then, we show that Triplet networks, (where state-of-the-art pretrained transformer-based models and modernized convolutional neural network architectures serve as the backbone and distinct topological features allow to represent support and query compounds), successfully identify the compounds with the desired properties. We also demonstrate that the applicability of topological feature maps can be successfully generalized to traditional ML algorithms such as random forests.

The distinct advantage of TDA tools, in particular persistent homology (PH), is that it enables effective integration of the domain information such as atomic mass, partial charge, bond type (single, double, triple, aromatic ring), ionization energy or electron affinity, which carry vital information regarding the chemical properties of a compound at multiple resolution levels during the graph filtration step. While common PH theory allows only one such domain function to be used in this process, with our novel multipersistence approach, we show it is possible to use more than one domain function. Topological fingerprints can effectively carry much finer chemical information of the compound structure informed by the multiple domain functions embedded in the process. Specifically, multiparameter persistence homology decomposes a $2D$ graph structure of a compound into a series of subgraphs using domain functions and generates hierarchical topological representations in multiple resolution levels. At each resolution stage, our algorithm sequentially generates finer topological fingerprints of the chemical substructures. We feed these topological fingerprints to suitable ML/DL methods, and our ToDD models achieve state-of-the-art in all benchmark datasets across all targets (See Table 1 and 2).

**The key contributions of this paper are:**

- We develop a transformative approach to generate molecular fingerprints. Using multipersistence, we produce highly expressive and unique topological fingerprints for compounds independent of scale and complexity. This offers a new way to describe and search chemical space relevant to both drug discovery and development.

- We bring a new perspective to multiparameter persistence in TDA and produce a computationally efficient multidimensional fingerprint of chemical data that can successfully incorporate more than one domain function to the PH process. These MP fingerprints harness the computational strength of linear representations and are suitable to be integrated

into a broad range of ML, DL, and statistical methods; and open a path for computationally efficient extraction of latent topological information.

- We prove that our multidimensional persistence fingerprints have the same important stability guarantees as the ones exhibited by the most currently existing summaries for single persistence.

- We perform extensive numerical experiments in VS, showing that our ToDD models outperform all state-of-the-art methods by a wide margin (See Figure 1).

## 2 Related Work

### 2.1 Virtual Screening

A key step in the early stages of the drug discovery process is to find active compounds that will be further optimized into potential drug candidates. One prevalent computational method that is widely used for compound prioritization with desired properties is *virtual screening* (VS). There are two major categories, i.e., structure-based virtual screening (SBVS) and ligand-based virtual screening (LBVS) [20]. SBVS uses the $3D$ structural information of both ligand (compound) and target protein as a complex [12, 52, 86]. SBVS methods generally require a good understanding of $3D$-structure of the target protein to explore the different poses of a compound in a binding pocket of the target. This makes the process computationally expensive. On the other hand, LBVS methods compare structural similarities of a library of compounds with a known active ligand [79, 69] with an underlying assumption that similar compounds are prone to exhibit similar biological activity. Unlike SBVS, LBVS only uses ligand information. The main idea is to produce effective fingerprints of the compounds and use ML tools to find similarities. Therefore, computationally less expensive LBVS methods can be more efficient with larger chemical datasets especially when the structure of the target receptor is not known [56].

In the last 3 decades, various LBVS methods have been developed with different approaches and these can be categorized into 3 classes depending on the fingerprint they produce: SMILES [81] and SMARTS [29] are examples of $1D$-methods which produce $1D$-fingerprints, compressing compound information to a vector. RASCAL [78], MOLPRINT2D [9], ECFP [80], CDK-graph [97],CDK-hybridization [85],SWISS [103], Klekota-Roth [53], MACSS [29], E-state [36] and SIMCOMP [37] are among $2D$ methods which uses $2D$-structure fingerprint and graph matching. Finally, examples of $3D$-methods are ROCS [38], USR [8], PatchSurfer [42] which use the $3D$-structure of compounds and their conformations ($3D$-position of the compound) [84]. On the other hand, while ML methods have been actively used in the field for the last two decades, new deep learning methods made a huge impact in drug discovery process in the last 5 years [88, 52, 82]. Further discussion of state-of-the-art ML/DL methods are given in Section 6 where we compare our models and benchmark against them.

### 2.2 Topological Data Analysis

TDA and tools of persistent homology (PH) have recently emerged as powerful approaches for ML, allowing us to extract complementary information on the observed objects, especially, from graph-structured data. In particular, PH has become popular for various ML tasks such as clustering, classification, and anomaly detection, with a wide range of applications including material science [68, 43], insurance [99, 46], finance [55], and cryptocurrency analytics [33, 4, 73]. (For more details see surveys [6, 22] and TDA applications library [34]) Furthermore, it has become a highly active research area to integrate PH methods into geometric deep learning (GDL) in recent years [41, 100, 19, 23]. Most recently, the emerging concepts of *multipersistence* (MP) are proposed to advance the success of single parameter persistence (SP) by allowing the use of more than one domain function in the process to produce more granular topological descriptors of the data. However, the MP theory is not sufficiently mature as it suffers from the nonexistence of the barcode decomposition relating to the partially ordered structure of the index set $\{(\alpha_i, \beta_j)\}$ [57, 89]. The existing approaches remedy this issue via slicing technique by studying one-dimensional fibers of the multiparameter domain [18], but choosing these directions suitably and computing restricted SP vectorizations are computationally costly which makes the approach inefficient in real life applications. There are several promising recent studies in this direction [11, 93, 24], but these approaches fail to provide a practical topological summary such as "multipersistence diagram", and an effective MP vectorization to be used in real life applications.

## 2.3 TDA in Virtual Screening

In [16, 15, 14], the authors obtained successful results by integrating single persistent homology outputs with various ML models. Furthermore, in [50], the authors used multipersistence homology with fibered barcode approach in the $3D$ setting and obtained promising results. In the past few years, TDA tools were also successfully combined with various deep learning models for SBVS and property prediction [71, 72]. In [66, 45, 61, 95, 62], the authors successfully used TDA methods to generate powerful molecular descriptors. Then, by using these descriptors, they highly boosted the performance of various ML/DL models and outperformed the existing models in several benchmark datasets. For a discussion and comparison of TDA techniques with other approaches in virtual screening and property prediction, see the review article [70]. In this paper, we follow a different approach and propose a framework by adapting multipersistence homology to VS process which produces fine topological fingerprints which are highly suitable for ML/DL methods.

# 3 Background

We first provide the necessary TDA background for our machinery. While our techniques are applicable to various forms of data, e.g., point clouds and images (for details, see Section B.2), here we focus on the graph setup in detail with the idea of mapping the atoms and bonds that make up a compound into a set of nodes and edges that represent an undirected graph.

## 3.1 Persistent Homology

Persistent homology (PH) is a key approach in TDA, allowing us to extract the evolution of subtler patterns in the data shape dynamics at multiple resolution scales which are not accessible with more conventional, non-topological methods [17]. In this part, we go over the basics of PH machinery on graph-structured data. For further background on PH, see Appendix A.1 and [27, 30].

For a given graph $\mathcal{G}$, consider a nested sequence of subgraphs $\mathcal{G}_1 \subseteq \ldots \subseteq \mathcal{G}_N = \mathcal{G}$. For each $\mathcal{G}_i$, define an abstract simplicial complex $\widehat{\mathcal{G}}_i$, $1 \leq i \leq N$, yielding a *filtration*, a nested sequence of simplicial complexes $\widehat{\mathcal{G}}_1 \subseteq \ldots \subseteq \widehat{\mathcal{G}}_N$. This step is crucial in the process as one can inject domain information to the machinery exactly at this step by using a filtering function from domain, e.g., atomic mass, partial charge, bond type, electron affinity, ionization energy (See Appendix A.1). After getting a filtration, one can systematically keep track of the evolution of topological patterns in the sequence of simplicial complexes $\{\widehat{\mathcal{G}}_i\}_{i=1}^N$. A $k$-dimensional topological feature (or $k$-hole) may represent connected components (0-dimension), loops (1-dimension) and cavities (2-dimension). For each $k$-dimensional topological feature $\sigma$, PH records its first appearance in the filtration sequence, say $\widehat{\mathcal{G}}_{b_\sigma}$, and first disappearence in later complexes, $\widehat{\mathcal{G}}_{d_\sigma}$ with a unique pair $(b_\sigma, d_\sigma)$, where $1 \leq b_\sigma < d_\sigma \leq N$. We call $b_\sigma$ *the birth time* of $\sigma$ and $d_\sigma$ *the death time* of $\sigma$. We call $d_\sigma - b_\sigma$ *the life span* (or persistence) of $\sigma$. PH records all these birth and death times of the topological features in *persistence diagrams*. Let $0 \leq k \leq D$ where $D$ is the highest dimension in the simplicial complex $\widehat{\mathcal{G}}_N$. Then $k^{th}$ persistence diagram $\mathrm{PD}_k(\mathcal{G}) = \{(b_\sigma, d_\sigma) \mid \sigma \in H_k(\widehat{\mathcal{G}}_i) \text{ for } b_\sigma \leq i < d_\sigma\}$. Here, $H_k(\widehat{\mathcal{G}}_i)$ represents the $k^{th}$ *homology group* of $\widehat{\mathcal{G}}_i$ which keeps the information of the $k$-holes in the simplicial complex $\widehat{\mathcal{G}}_i$. Most common dimensions used in practice are 0 and 1, i.e., $PD_0(\mathcal{G})$ and $PD_1(\mathcal{G})$. For sake of notations, further we skip the dimension (subscript $k$). With the intuition that the topological features with long life spans (persistent features) describe the hidden shape patterns in the data, these persistence diagrams provide a unique topological fingerprint of $\mathcal{G}$. We give the further details of the PH machinery and how to integrate domain information into the process in Appendix A.1.

## 3.2 Multidimensional Persistence

MultiPersistence (MP) significantly boosts the performance of the single parameter persistence technique described in Appendix A.1. The reason for the term "single" is that we are filtering the data in only one direction $\mathcal{G}_1 \subset \cdots \subset \mathcal{G}_N = \mathcal{G}$. As explained in Appendix A.1, the construction of the filtration is the key step to inject domain information to process and to find the hidden patterns of the data. If one uses a function $f : \mathcal{V} \to \mathbb{R}$ which has valuable domain information, then this induces a single parameter filtration as above. However, various data have more than one domain function

to analyze the data, and using them simultaneously would give a much better understanding of the hidden patterns. For example, if we have two functions $f, g : \mathcal{V} \to \mathbb{R}$ (e.g., atomic mass and partial charge) with valuable complementary information of the network (compound), MP idea is presumed to produce a unique topological fingerprint combining the information from both functions. These pair of functions $f, g$ induces a multivariate filtering function $F : \mathcal{V} \to \mathbb{R}^2$ with $F(v) = (f(v), g(v))$. Again, one can define a set of nondecreasing thresholds $\{\alpha_i\}_1^m$ and $\{\beta_j\}_1^n$ for $f$ and $g$ respectively. Let $\mathcal{V}_{ij} = \{v_r \in \mathcal{V} \mid f(v_r) \leq \alpha_i, g(v_r) \leq \beta_j\}$, i.e., $\mathcal{V}_{ij} = F(v_r) \preceq (\alpha_i, \beta_j)$. Define $\mathcal{G}_{ij}$ to be the induced subgraph of $\mathcal{G}$ by $\mathcal{V}_{ij}$, i.e., the smallest subgraph of $\mathcal{G}$ generated by $\mathcal{V}_{ij}$. Then, instead of a single filtration of complexes $\{\widehat{\mathcal{G}}_i\}$, we get a *bifiltration* of complexes $\{\widehat{\mathcal{G}}_{ij} \mid 1 \leq i \leq m, 1 \leq j \leq n\}$ which is a $m \times n$ rectangular grid of simplicial complexes. Again, the MP idea is to keep track of the $k$-dimensional topological features in this grid $\{\widehat{\mathcal{G}}_{ij}\}$ by using the corresponding homology groups $\{H_k(\widehat{\mathcal{G}}_{ij})\}$ (MP module).

As noted in Section 2, because of the technical problems related to partially ordered structure of the MP module, the MP theory has no sound definition yet (e.g., birth/death time of a topological feature in MP grid), and there is no effective way to facilitate this promising idea in real life applications. In the following, we overcome this problem by producing highly effective fingerprints by utilizing the *slicing* idea in the MP grid in a structured way.

## 4 New Topological Fingerprints of the Compounds with Multipersistence

ToDD framework produces fingerprints of compounds as multidimensional vectors by expanding single persistence (SP) fingerprints (Appendix A.1). While our construction is applicable and suitable for various forms of data, here we focus on graphs, and in particular, compounds for virtual screening. We obtain a $2D$ matrix (or 3D array) for each compound as its fingerprint employing 2 or 3 functions/weights (e.g., atomic mass, partial charge, bond type, electron affinity, ionization energy) to perform graph filtration. We explain how to generalize our framework to other types of data in Appendix B.2. In Appendix B.4, we construct the explicit examples of MP Fingerprints for most popular SP Vectorizations, e.g., Betti, Silhouette, Landscapes.

Our framework basically expands a given SP vectorization to a multidimensional vector by utilizing MP approach. In technical terms, by using the existing SP vectorizations, we produce multidimensional vectors by effectively using one of the filtering direction as *slicing direction* in the multipersistence module. We explain our process in three steps.

*Step 1 - Bifiltrations:* This step basically corresponds to obtaining relevant *substructures* from the given compound in an organized way. Here, we give the computationally most feasible method, called *sublevel bifiltration* with 2 functions. Depending on the task and dataset, the other filtration types or more functions/weights can be more useful. In Section B.5, we give details for other filtration methods we use in our experiments. i.e., Vietoris-Rips (distance) and weight filtration.

Let $f, g : \mathcal{V} \to \mathbb{R}$ be two filtering functions with threshold sets $\{\alpha_i\}_{i=1}^m$ and $\{\beta_j\}_{j=1}^n$ respectively (e.g., $f$ is atomic mass, and $g$ is partial charge). Let $\mathcal{V}_i = \{v_r \in \mathcal{V} \mid f(v_r) \leq \alpha_i\}$ and let $\mathcal{G}_i$ be the induced subgraph of $\mathcal{G}$ by $\mathcal{V}_i$, i.e. add any edge in $\mathcal{G}$ whose endpoints are in $\mathcal{V}_i$. Similarly, let $\mathcal{V}_{ij} = \{v_r \in \mathcal{V} \mid f(v_r) \leq \alpha_i$ and $g(v_r) \leq \beta_j\} \subset \mathcal{V}_i$. Let $\mathcal{G}_{ij}$ be the induced subgraph of $\mathcal{G}_i$ by $\mathcal{V}_{ij}$. Then, define $\widehat{\mathcal{G}}_{ij}$ as *the clique complex* of $\mathcal{G}_{ij}$ (See Section A.1). In particular, by using the first function ($f$), we filter $\mathcal{G}$ in one (say vertical) direction $\{\mathcal{G}_i\}$. Then, by using the second function ($g$), we filter each $\mathcal{G}_i$ in horizontal direction and obtain a bifiltration $\{\widehat{\mathcal{G}}_{ij}\}$. These subgraphs $\{\mathcal{G}_{ij}\}$ represent the induced substructures of the compound $\mathcal{G}$ by using the filtering functions $f$ and $g$.

In Figure 2 and 3, we give an example of sublevel bifiltration of the compound cytosine by atomic number and partial charge functions. In Figure 2, atom types are coded by their color. Atomic numbers are given in the parenthesis. White=Hydrogen (1), Gray=Carbon (6), Blue=Nitrogen (7), and Red=Oxygen (8). The decimal numbers next to atoms represent their partial charges.

*Step 2 - Persistence Diagrams:* After constructing the bifiltration $\widehat{\mathcal{G}}_{ij}$, the second step is to obtain persistence diagrams for each row. By restricting the bifiltration to a single row, for each $1 \leq i_0 \leq m$, one obtains a single filtration $\widehat{\mathcal{G}}_{i_0 1} \subseteq \widehat{\mathcal{G}}_{i_0 2} \ldots \subseteq \widehat{\mathcal{G}}_{i_0 n}$ in horizontal direction. This is called a *horizontal slice* in the bipersistence module. Each such single filtration induces a persistence diagram $PD(\mathcal{G}_i) = \{(b_j, d_j) \mid 0 \leq b_j < d_j \leq n\}$. This produces $m$ persistence diagrams $\{PD(\mathcal{G}_i)\}$. Notice

that one can consider $PD(\mathcal{G}_i)$ as the single persistence diagram of the "substructure" $\mathcal{G}_i$ filtered by the second function $g$ (See Section A.1).

*Step 3 - Vectorization:* The final step is to use a vectorization on these $m$ persistence diagrams. Let $\varphi$ be a single persistence vectorization, e.g., Betti, Silhouette, Entropy, Persistence Landscape or Persistence Image. Specifically, we use Betti to ease computational complexity. By applying the chosen SP vectorization $\varphi$ to each PD, we obtain a function $\varphi_i = \varphi(PD(\mathcal{G}_i))$ where in most cases it is a single variable function on the threshold domain $[0, n]$, i.e., $\varphi_i : [1, n] \to \mathbb{R}$. The number of thresholds $m, n$ are important as it determines the size of our topological fingerprint. As most such vectorizations are induced from a discrete set of points $PD(\mathcal{G})$, it is common to express them as vector in the form $\vec{\varphi} = [\varphi(1)\ \varphi(2)\ \ldots\ \varphi(n)]$. In the examples in Section B.4, we explain this conversion explicitly for different vectorizations. Hence, we obtain a vector $\vec{\varphi}_i$ of size $1 \times n$ for each row $1 \le i \le m$.

Now, we can define our topological fingerprint $\mathbf{M}_\varphi$ which is a $2D$-vector (a matrix)

$$\mathbf{M}_\varphi^i = \vec{\varphi}_i \quad \text{for} \quad 1 \le i \le m,$$

where $\mathbf{M}_\varphi^i$ is the $i^{th}$-row of $\mathbf{M}_\varphi$. Hence, $\mathbf{M}_\varphi$ is a $2D$-vector of size $m \times n$. Each row $\mathbf{M}_\varphi^i$ is the vectorization of the persistence diagram $PD(\mathcal{G}_i)$ via the SP vectorization method $\varphi$. We use the first filtering function $f$ to get a finer look at the graph as it defines the subgraphs $\mathcal{G}_1 \subseteq \ldots \subseteq \mathcal{G}_m = \mathcal{G}$. Then, by using the second function $g$ on each $\mathcal{G}_i$, we record the evolution of topological features in each $\mathcal{G}_i$ as $PD(\mathcal{G}_i)$. While this construction gives our $2D$ (matrix) fingerprints $\mathbf{M}_\varphi$, one can also use 3 functions/weights for filtration and obtain a finer $3D$ (array) topological fingerprint (Section B.3).

In a way, we look at $\mathcal{G}$ with a $2D$ resolution (functions $f$ and $g$ as lenses) and keep track of the evolution of topological features in the induced substructures $\{\mathcal{G}_{ij}\}$. The main advantage of this technique is that the outputs are fixed size multidimensional vectors for each dataset which are suitable for various ML/DL models.

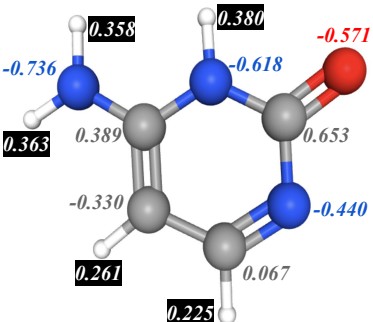

Figure 2: **Cytosine**. Atom types are coded by their color: White=Hydrogen, Gray=Carbon, Blue=Nitrogen, and Red=Oxygen. The decimal numbers next to atoms represent their partial charges.

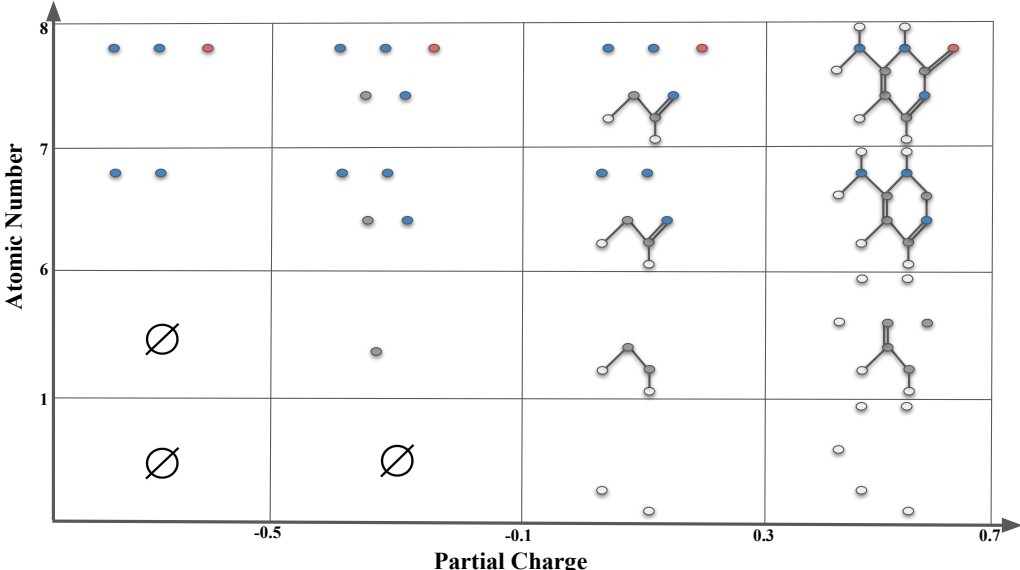

Figure 3: **Sublevel bifiltration of cytosine** is induced by filtering functions atomic charge $f$ and atomic number $g$. In the horizontal direction, thresholds $\alpha = -0.5, -0.1, +0.3, +0.7$ filters the compound into substructures $f(v) \le \alpha$ with respect to their partial charge. In the vertical direction, thresholds $\beta = 1, 6, 7, 8$ filters the compound in the substructures $g(v) \le \beta$ with respect to atomic numbers. Each box $\Delta_{\alpha,\beta}$ indexed by their upper right coordinates $(\alpha, \beta)$ representing the substructure $\Gamma_{\alpha,\beta} = \{f(v) \le \alpha, g(v) \le \beta\}$. Whenever two nodes (atoms) are in the substructure, if there is an edge (bond) between them in the original compound, we include the edge in the substructure.

### 4.1 Stability of the MP Fingerprints

We further show that when the source single parameter vectorization $\varphi$ is stable, then so is its induced MP Fingerprint $\mathbf{M}_\varphi$. (We give the details of stability notion in persistence theory and proof of the following theorem in Section B.1.)

**Theorem:** *Let $\varphi$ be a stable SP vectorization. Then, the induced MP Fingerprint $\mathbf{M}_\varphi$ is also stable, i.e., with the notation introduced in Section B.1, there exists $\widehat{C}_\varphi > 0$ such that for any pair of graphs $\mathcal{G}^+$ and $\mathcal{G}^-$, we have the following inequality.*

$$\mathfrak{D}(\mathbf{M}_\varphi(\mathcal{G}^+), \mathbf{M}_\varphi(\mathcal{G}^-)) \leq \widehat{C}_\varphi \cdot \mathbf{D}_{p_\varphi}(\{PD(\mathcal{G}^+)\}, \{PD(\mathcal{G}^-)\})$$

## 5 Datasets

**Cleves-Jain:** This is a relatively small dataset [26] that has 1149 compounds.[*] There are 22 different drug targets, and for each one of them the dataset provides only 2-3 template active compounds dedicated for model training, which presents a few-shot learning task. All targets $\{q\}$ are associated with 4 to 30 active compounds $\{L_q\}$ dedicated for model testing. Additionally, the dataset contains 850 decoy compounds ($D$). The aim is for each target $q$, by using the templates, to find the actives $L_q$ among the pool combined with decoys $L_q \cup D$, i.e., same decoy set $D$ is used for all targets.

**DUD-E Diverse:** DUD-E (Directory of Useful Decoys, Enhanced) dataset [67] is a comprehensive ligand dataset with 102 targets and approximately 1.5 million compounds.[*] The targets are categorized into 7 classes with respect to their protein type. The "Diverse subset" of DUD-E contains targets from each category to give a balanced benchmark dataset for VS methods. Diverse subset contains 116,105 compounds from 8 target and 8 decoy sets. One decoy set is used per target.

More detailed information about each dataset can be found in Appendix C.1.

## 6 Experiments

### 6.1 Setup

**Macro Design** We construct different ToDD (Topological Drug Discovery) models, namely ToDD-ViT, ToDD-ConvNeXt and ToDD-RF to test the generalizability and scalability of topological features while employing different ML models and training datasets of various sizes. Many neural network architectural choices and ML models can be incorporated in our ToDD method. ToDD-ViT and ToDD-ConvNeXt are Triplet network architectures with Vision Transformer (ViT_b_16) [28] and ConvNeXt_tiny models [63], pretrained on ILSVRC-2012 ImageNet, serving as the backbone of the Triplet network. MP signatures of compounds are applied nearest neighbour interpolation to increase their resolutions to $224^2$, followed by normalization. We only use GaussianBlur with kernel size $5^2$ and standard deviation 0.05 as a data augmentation technique. Transfer learning via fine-tuning ViT_b_16 and ConvNeXt_tiny models using Adam optimizer with a learning rate of 5e-4, no warmup or layerwise learning rate decay, cosine annealing schedule for 5 epochs, stochastic weight averaging for 5 epochs, weight decay of 1e-4, and a batch size of 64 for 10 epochs in total led to significantly better performance in Enrichment Factor and ROC-AUC scores compared to training from scratch. The performance of all models was assessed by 5-fold cross-validation (CV).

Due to structural isomerism, molecules with identical molecular formulae can have the same bonds, but the relative positions of the atoms differ [76]. ViT has much less inductive bias than CNNs, because locality and translation equivariance are embedded into each layer throughout the entire network in CNNs, whereas in ViT self-attention layers are global and only MLP layers are translationally equivariant and local [28]. Hence, ViT is more robust to distinct arrangements of atoms in space, also referred to as molecular conformation. On a small-scale dataset like Cleves-Jain, ViT exhibits impressive performance. However, the memory and computational costs of dot-product attention blocks of ViT grow quadratically with respect to the size of input, which limits its application on large-scale datasets [60, 83]. Another major caveat is that the number of triplets grows cubically with

---

[*]Cleves-Jain dataset: `https://www.jainlab.org/Public/SF-Test-Data-DrugSpace-2006.zip`
[*]DUD-E Diverse dataset: `http://dude.docking.org/subsets/diverse`

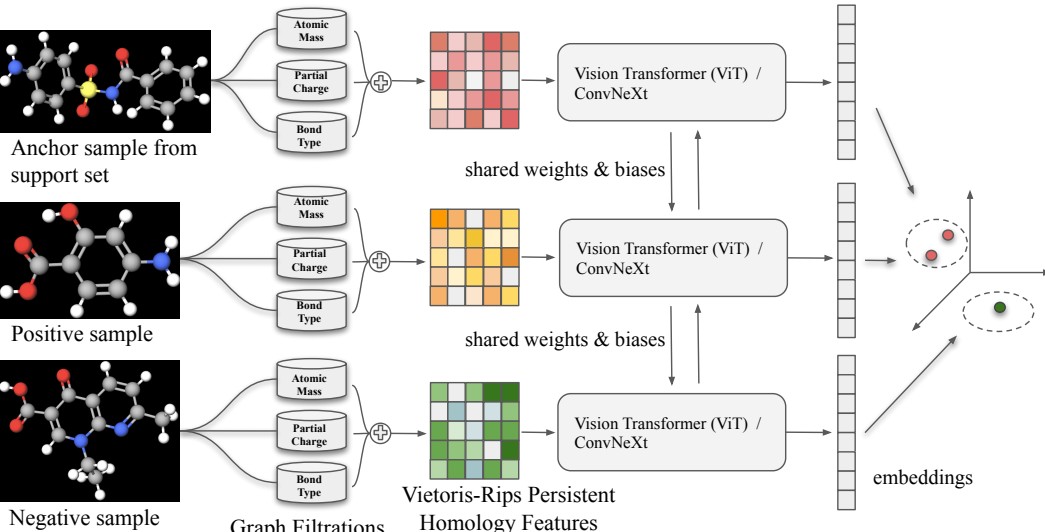

**Figure 4: End-to-end model pipeline.** Anchor sample, $x$, and positive sample, $x^+$, are compounds that can bind to the same drug target, whereas negative sample, $x^-$, is a decoy. $2D$ graph representation of each compound is decomposed into subgraphs induced by the periodic properties: atomic mass, partial charge and bond type. Potentially these domain functions can be augmented using other periodic properties such as ionization energy and electron affinity as well as using molecular information such as chirality, orbital hybridization, number of Hydrogen bonds or number of conjugated bonds at the cost of computational complexity. Subgraphs may have isolated nodes and edges. Our MP framework establishes Vietoris-Rips complexes for each subgraph and provides MP signatures (topological fingerprints) of the compounds. Both ToDD-ViT and ToDD-ConvNeXt can encode the pair of distances between a positive query and a negative query against an anchor sample from the support set.

the size of the dataset. Since ConvNeXt depends on a fully-convolutional paradigm, its inherently efficient design is viable on large-scale datasets like DUD-E Diverse. As depicted in Figure 4, ToDD-ViT and ToDD-ConvNeXt project semantically similar MP signatures of compounds from data manifold onto metrically close embeddings using triplet margin loss with margin $\alpha = 1.0$ and norm $p = 2$ as provided in Equation 1. Analogously, semantically different MP signatures are projected onto metrically distant embeddings.

$$L(x, x^+, x^-) = \max(0, \alpha + \|\mathbf{f}(\mathbf{x}) - \mathbf{f}(\mathbf{x^+})\|_p - \|\mathbf{f}(\mathbf{x}) - \mathbf{f}(\mathbf{x^-})\|_p) \tag{1}$$

**Sampling Strategy** Learning metric embeddings via triplet margin loss on large-scale datasets poses a special challenge in sampling all distinct triplets $(x, x^+, x^-)$, and collecting them into a single database causes excessive overhead in computation time and memory. Let $P$ be a set of compounds, $x_i$ denotes a compound that inhibits the drug target $i$, and $d_{ij} = d(x_i, x_j) \in \mathbb{R}$ denotes a pairwise distance measure which estimates how strongly $x_i \in P$ is similar to $x_j \in P$. The distance metric can be chosen as Euclidean distance, cosine similarity or dot-product between embedding vectors. We use pairwise Euclidean distance computed by the pretrained networks in the implementation. Since triplets $(x, x^+, x^-)$ with $d(x, x^-) > d(x, x^+) + \alpha$ have already negative queries sufficiently distant to the anchor compounds from the support set in the embedding space, they are not sampled to create the training dataset. We only sample triplets that satisfy $d(x, x^-) < d(x, x^+)$ (where negative query is closer to the anchor than the positive) and $d(x, x^+) < d(x, x^-) < d(x, x^+) + \alpha$ (where negative query is more distant to the anchor than the positive, but the distance is less than the margin).

**Enrichment Factor** (EF) is the most common performance evaluation metric for VS methods [90]. VS method $\varphi$ ranks compounds in the database by their similarity scores. We measure the similarity score using the inverse of Euclidean distance between the embeddings of an anchor and drug candidate. Let $N$ be the total number of ligands in the dataset, $A_\varphi$ be the number of true positives (i.e., correctly predicted active ligands) ranked among the top $\alpha\%$ of all ligands ($N_\alpha = N \cdot \alpha\%$) and $N_{\text{actives}}$ be the number of active ligands in the whole dataset. Then, $EF_{\alpha\%} = \frac{A\varphi/N_{\text{actives}}}{\alpha/100}$. In other words, $EF_{\alpha\%}$ interprets as how much VS method $\varphi$ *enrich* the possibility of finding active ligand in the first $\alpha\%$ of all ligands with respect to the random guess. This method is also known as *precision at k* in the literature. With this definition, the max score for $EF_{\alpha\%}$ is $\frac{100}{\alpha}$, i.e., 100 for $EF_{1\%}$ and 20 for $EF_{5\%}$.

## 6.2 Experimental Results

We compare our methods against the 23 state-of-the-art baselines (see Appendix C.2).

Table 1: Comparison of EF 2%, 5%, 10% and ROC-AUC values between ToDD and other virtual screening methods on the Cleves-Jain dataset.

| Model | EF 2% (max. 50) | EF 5% (max. 20) | EF 10% (max. 10) | ROC-AUC |
|---|---|---|---|---|
| USR [8] | 10.0 | 6.2 | 4.1 | 0.76 |
| GZD [92] | 13.4 | 8.0 | 5.3 | 0.81 |
| PS [42] | 10.7 | 6.6 | 4.9 | 0.78 |
| ROCS [38] | 20.1 | 10.7 | 6.2 | _0.83_ |
| USR + GZD [84] | 13.7 | 7.7 | 4.7 | 0.81 |
| USR + PS [84] | 13.1 | 7.9 | 5.0 | 0.80 |
| USR + ROCS [84] | 17.1 | 9.1 | 5.4 | _0.83_ |
| GZD + PS [84] | 16.0 | 9.1 | 5.9 | 0.82 |
| PH_VS [50] | 18.6 | NA | NA | NA |
| GZD + ROCS [84] | 20.3 | _10.8_ | 5.3 | _0.83_ |
| PS + ROCS [84] | _20.5_ | 10.7 | _6.4_ | _0.83_ |
| **ToDD-RF** | 35.2±2.3 | 15.6±1.0 | 8.1±0.4 | **0.94**±0.02 |
| **ToDD-ViT** | **39.6**±1.4 | **18.6**±0.4 | **9.9**±0.1 | 0.90±0.01 |
| Relative gains | 92.9% | 83.7% | 54.1% | 13.3% |

Table 2: Comparison of EF 1% (max. 100) between ToDD and other virtual screening methods on 8 targets of the DUD-E Diverse subset.

| Model | AMPC | CXCR4 | KIF11 | CP3A4 | GCR | AKT1 | HIVRT | HIVPR | Avg. |
|---|---|---|---|---|---|---|---|---|---|
| Findsite [101] | 0.0 | 0.0 | 0.9 | 21.7 | 34.2 | 39.0 | 1.2 | 34.7 | 16.5 |
| Fragsite [102] | 4.2 | 42.5 | 0.0 | 32.9 | 29.1 | 47.1 | 2.4 | 48.7 | 25.9 |
| Gnina [87] | 2.1 | 15.0 | 38.0 | 1.2 | 39.0 | 4.1 | 11.0 | 28.0 | 17.3 |
| GOLD-EATL [96] | 25.8 | 20.0 | 33.5 | 17.9 | 34.6 | 29.2 | 28.7 | 23.4 | 26.6 |
| Glide-EATL [96] | 35.5 | 20.8 | 30.5 | 15.1 | 24.0 | 31.6 | 29.0 | 22.0 | 26.1 |
| CompM [96] | 32.3 | 25.0 | 35.5 | 33.6 | 37.1 | 44.2 | 30.2 | 25.0 | 32.9 |
| CompScore [75] | _39.6_ | 51.6 | 51.3 | 14.0 | 27.1 | 37.6 | 21.8 | 18.2 | 32.7 |
| CNN [77] | 2.1 | 5.0 | 11.2 | 28.7 | 12.8 | 84.6 | 12.2 | 9.9 | 20.8 |
| DenseFS [44] | 14.6 | 5.0 | 4.3 | _44.3_ | 20.9 | _89.4_ | 12.8 | 8.4 | 25.0 |
| SIEVE-Score [98] | 30.7 | _61.1_ | 53.4 | 6.7 | 33.3 | 42.1 | 39.8 | 38.3 | 38.2 |
| DeepScore [94] | 28.1 | 56.8 | _54.3_ | 37.1 | _40.9_ | 59.0 | _43.8_ | 62.8 | _47.9_ |
| RF-Score-VSv3 [98] | 32.3 | 60.9 | 4.5 | 25.9 | 32.5 | 41.9 | 39.8 | _65.7_ | 37.9 |
| **ToDD-RF** | 42.9±4.5 | **92.3**±3.2 | **75.0**±5.0 | **67.6**±3.4 | **78.9**±4.0 | **90.7**±1.3 | **64.1**±2.3 | **92.1**±1.5 | **73.7** |
| **ToDD-ConvNeXt** | **46.2**±3.6 | 84.6±2.8 | 72.5±3.6 | 28.8±2.8 | 46.0±2.0 | 81.2±2.5 | 37.5±3.6 | 74.6±1.0 | 58.9 |
| Relative gains | 16.7% | 51.1% | 38.1% | 52.6% | 92.9% | 1.5% | 46.3% | 40.2% | 53.9% |

Relative gains are relative to the next best performing model. Based on the results (mean and standard deviation of EF scores evaluated by CV) reported in Table 1 and 2, we observe the following:

- ToDD models consistently achieve the best performance on both Cleves-Jain and DUD-E Diverse datasets across all targets and $EF_{\alpha\%}$ levels.

- ToDD learns hierarchical topological representations of compounds using their atoms' periodic properties, and captures the complex chemical properties essential for high-throughput VS. These strong hierarchical topological representations enable ToDD to become a model-agnostic method that is extensible to state-of-the-art neural networks as well as ensemble methods like random forests (RF).

- For small-scale datasets such as Cleves-Jain, RF is less accurate than ViT despite regularization by bootstrapping and using pruned, shallow trees, because small variations in the data may generate significantly different decision trees. For large-scale datasets such as DUD-E Diverse, ToDD-RF and ToDD-ConvNeXt exhibit comparable performances except for: CP3A4, GCR and HIVRT. We conclude that transformer-based models are more robust than convolutional models and RF classifiers despite increased computation time.

## 6.3 Computational Complexity

Computational complexity (CC) of MP Fingerprint $\mathbf{M}_\psi^d$ depends on the vectorization $\psi$ used and the number $d$ of the filtering functions one uses. CC for a single persistence diagram $PD_k$ is $\mathcal{O}(\mathcal{N}^3)$ [74],

where $\mathcal{N}$ is the number of $k$-simplices. If $r$ is the resolution size of the multipersistence grid, then $CC(\mathbf{M}_\psi^d) = \mathcal{O}(r^d \cdot \mathcal{N}^3 \cdot C_\psi(m))$ where $C_\psi(m)$ is CC for $\psi$ and $m$ is the number of barcodes in $PD_k$, e.g., if $\psi$ is Persistence Landscape, then $C_\psi(m) = m^2$ [13] and hence CC for MP Landscape with three filtering functions ($d = 3$) is $\mathcal{O}(r^3 \cdot \mathcal{N}^3 \cdot m^2)$. On the other hand, for MP Betti summaries, one does not need to compute persistence diagrams, but the rank of homology groups in the MP module. Hence, for MP Betti summary, the computational complexity is indeed much lower by using minimal representations [58, 51]. To expedite the execution time, the feature extraction task is distributed across the 8 cores of an Intel Core i7 CPU (100GB RAM) running in a multiprocessing process. See Appendix C.4 for an additional analysis of computation time to extract MP fingerprints from the datasets. Furthermore, all ToDD models require substantially fewer computational resources during training compared to current graph-based models that encode a compound through mining common molecular fragments, a.k.a., motifs [47]. Training time of ToDD-ViT and ToDD-ConvNeXt for each individual drug target takes less than 1 hour on a single GPU (NVIDIA RTX 2080 Ti).

### 6.4 Ablation Study

We tested a number of ablations of our model to analyze the effect of its individual components and to further investigate the effectiveness of our topological fingerprints.

**Multimodal Learning** We first address the question of how adding different domain information improves the model performance. In Appendix C.3, we demonstrate one-by-one the importance of each modality (atomic mass, partial charge and bond type) used for graph filtration to the classification of each target. We find that their importance varies across targets in a unimodal setting, but the orthogonality of these information sources offers significant gain in EF scores when the MP signatures learned from each modality are integrated into a joined multimodal representation. Tables 5, 6, 7 and 8 provide detailed results for the performance of each modality across all drug targets.

**Morgan Fingerprints** We quantitatively analyze the explainability of our models' success by replacing topological fingerprints computed via multiparameter persistence with the most popular fingerprinting method: Morgan fingerprints. Our results in Appendix C.5 show that ToDD engineers features that represent the underlying attributes of compounds significantly better than the Morgan algorithm to identify the active compounds across all drug targets. We provide detailed tabulated results of our benchmarking study across all drug targets in Tables 10 and 11.

**Network Architecture** We investigated ways to leverage deep metric learning by architecting $i$) a Siamese network trained with contrastive loss, $ii$) a Triplet network trained with triplet margin loss, and $iii$) a Triplet network trained with circle loss. Based on our preliminary experiments, the embeddings learned by $i$ and $iii$ provide sub-par results for compound classification, hence we use $ii$.

## 7 Conclusion

We have proposed a new idea of the topological fingerprints in VS, allowing for deeper insights into structural organization of chemical compounds. We have evaluated the predictive performance of our ToDD methodology for computer aided drug discovery on benchmark datasets. Moreover, we have demonstrated that our topological descriptors are model-agnostic and have proven to be exceedingly competitive, yielding state-of-the-art results unequivocally over all baselines. A future research direction is to enrich ToDD with different VS modalities, and use it on ultra-large virtual compound libraries. It is important to note that this new way of capturing the chemical information of compounds provides a transformative perspective to every level of the pharmaceutical pipeline from the very early phases of drug discovery to the final stages of formulation in development.

## 8 Acknowledgments

YG, BC, YC and ISD were partially supported by Simons Collaboration Grant # 579977, the National Science Foundation (NSF) under award # ECCS 2039701, the Department of the Navy, Office of Naval Research (ONR) under ONR award # N00014-21-1-2530. Part of YG's contribution is also based upon work supported by (while serving at) the NSF. Any opinions, findings, and conclusions or recommendations expressed in this material are those of the author(s) and do not necessarily reflect the views of the NSF and/or the ONR.

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
