# OpenReview forum: "ToDD: Topological Compound Fingerprinting in Computer-Aided Drug Discovery"
_NeurIPS.cc/2022/Conference — NeurIPS 2022 Accept_

### Official Review · Reviewer_ACvS · 2022-07-11

**Rating:** 9
**Confidence:** 4
**Soundness:** 3 good
**Presentation:** 4 excellent
**Contribution:** 4 excellent

**Summary:**

In this paper, the authors develop multiparameter persistence (MP) homology based topological fingerprints for virtual screening. In their model, a compound is decomposed into chemical substructures at different scales and their topological information is extracted by using MP homology features. These topological fingerprints are combined with learning models. Their models are extensively tested on two benchmark datasets, i.e., Cleves-Jain and DUD-E Diverse dataset. It has been found that their models have better performance than existing models.

**Questions:**

The major issue is the missing of important literature in the related work. Even though the authors have mentioned some TDA-based drug design models, i.e., refs 13-15, they have been only very briefly mentioned in the supplementary information. Further, these works are from 2017 to 2018. In fact, recently, there are great progresses in TDA-based drug design (and virtual screening). The authors should discussed them in the related works.

For instance,

1) Duc Duy Nguyen, Zixuan Cang, Kedi Wu, Menglun Wang, Yin Cao  and Guo-Wei Wei, Mathematical deep learning for pose and binding affinity prediction and ranking in D3R Grand Challenges, Journal of Computer Aided Molecular Design, 33, 71-82   (2019).

2) Duc Nguyen, Zixuan Cang, and Guo-Wei Wei, A review of mathematical representations of biomolecular data, Physical Chemistry Chemical Physics, 22, 4343-4367 (2020).

3) Duc Duy Nguyen, Kaifu Gao, Menglun Wang, and Guo-Wei Wei, MathDL: Mathematical deep learning for D3R Grand Challenge 4, Journal of Computer Aided Molecular Design,  34, 131-147 (2020).

4) Zhenyu Meng and Kelin Xia, "Persistent spectral–based machine learning (PerSpect ML) for protein-ligand binding affinity prediction." Science Advances, 7 (19), eabc5329 (2021)

5) Peiran Jiang, Ying Chi, Xiao-Shuang Li, Xiang Liu, Xian-Sheng Hua, and Kelin Xia, "Molecular persistent spectral image (Mol-PSI) representation for machine learning models in drug design." Briefings in Bioinformatics, 23 (1), bbab527 (2022)

6) Xiang Liu, Huitao Feng, Jie Wu, and Kelin Xia, "Dowker complex based machine learning (DCML) models for protein-ligand binding affinity prediction." PLOS Computational Biology, 18(4), e1009943 (2022)

7) Xiang Liu, and Kelin Xia, "Persistent Tor-algebra based stacking ensemble learning (PTA-SEL) for protein-protein binding affinity prediction", ICLR 2022 Workshop on Geometrical and Topological Representation Learning (2022)

8) Xiang Liu, and Kelin Xia, "Neighborhood complex based machine learning (NCML) models for drug design." In Interpretability of Machine Intelligence in Medical Image Computing, and Topological Data Analysis and Its Applications for Medical Data, pp. 87-97. Springer, Cham (2021).


**Ethics Review Area:**

["I don’t know"]

**Limitations:**

None.

**Strengths And Weaknesses:**

The models are innovative and the results are great! However, TDA-based learning models for drug design (including virtual screening) has been a hot research areas for the past decade. There is a great amount of works in this area that is missing in the current paper.

---

> ### Author Response · Authors · 2022-08-01
> **Authors' Response to Reviewer ACvS**
>
> We thank the reviewer for appreciating the novelty of this interdisciplinary approach and for the valuable feedback and references.  It has been a rewarding but challenging process to bring domain experts together from vastly different disciplines including mathematics - topology, applied statistics, engineering, physics, drug discovery & development for this collaboration. We are very much encouraged to see that the reviewer found our work exciting and results promising.
>
> Per your suggestions, we made a major revision in the Related Work section (Section 2.3). Specifically, we include all the references suggested, and discuss the relevance of TDA in molecular data analysis and protein-ligand binding affinity prediction to accelerate drug discovery for a more detailed review.
>
> In general, we hope that TDA concepts, as outlined in the papers listed by the Reviewer and the one presented here, will contribute to shaping a new research direction in the ML community on topological ML for drug discovery.
>
> Thank you very much again for your suggestions to improve our paper. Please do not hesitate to post further comments or questions.

---

> > ### Comment · Reviewer_ACvS · 2022-08-08
> > **All my questions are well addressed**
> >
> > All my questions are well addressed. I think the paper is of great interests and the results are very convincing!

---

> > > ### Author Response · Authors · 2022-08-08
> > > **Thank you very much!**
> > >
> > > We are very grateful for the constructive and motivating feedback and for recognizing the importance and novelty of this project on TDA for drug discovery!

---

### Official Review · Reviewer_WV81 · 2022-07-11

**Rating:** 4
**Confidence:** 4
**Soundness:** 2 fair
**Presentation:** 2 fair
**Contribution:** 2 fair

**Summary:**


The authors proposed a new topological fingerprint in virtual screening to understand structural organization of chemical compounds. The method is based on multiparameter persistence homology, which produces topological fingerprints of the compounds as multidimensional vectors. They show that the proposed topological descriptors outperform the state-of-the-art methods on benchmark datasets.


**Questions:**

The authors should use popular performance measures such as AUC scores of ROC curve and precision-recall curve. I'm afraid that researchers in the NeurIPS community are not familiar with the performance measures used in this paper (e.g., enrichment factor alpha %).

There are many fingerprints and descriptors in chemoinformatics. Examples include the following papers:
FCFP (Rogers et al, J. Chem. Inf. Model., 2010)
CDK graph (Yap et al, J. Comput. Chem., 2011)
CDK hybridization (Steinbeck et al, Curr. Pharm. Des., 2006)
Klekota-Roth (Klekota et al, Bioinformatics, 2008)
MACCS (Durant et al, J. Chem. Inf. Comp. Sci., 2002)
PubChem (Chen et al, J. Chem. Inf. Model., 2009)
E-state (Hall et al, J. Chem. Inf. Comp. Sci., 1995)
The proposed fingerprint is better than the other fingerprints?

The prediction accuracy in virtual screening depends too much on the number of positive examples (active compounds) in the training set. In my experience, transformer-based methods do not work well when there are few positive examples in the training set.

It is hard to follow this paper.
The explanations on the detailed methods are not clear.

The output of the propose method is not clear.
What is the dimension/size of the proposed topological fingerprint?

The number of compounds in the chemical library is often huge. How is the scalability of the proposed method?


**Limitations:**

The paper does not discuss the limitations of the proposed method clearly.
The performance is not rigorously verified.


**Strengths And Weaknesses:**

Fingerprinting is an important research topic in drug discovery.
The proposed method provided high accuracy.

There are many previous works on chemical fingerprints and descriptors, but the relationship with related works is not clear.
The authors did not compare the performance with many other fingerprints.

---

> ### Author Response · Authors · 2022-08-01
> **Authors' Response to Reviewer WV81 on Questions 1-2**
>
> We thank the reviewer for the time and effort spent in reviewing our paper, and for the detailed suggestions. Reflecting your valuable comments, we made a major revision adding multiple figures in Appendix Section B.5 to facilitate the understanding of graph filtration and computation of multiparameter (MP) persistence using Vietoris-Rips (VR) simplicial complexes. We believe that our paper was significantly improved, thanks to your comments. We hope that our responses would be sufficiently informative for you to reconsider assessing a higher rating for the revised paper.
>
> Detailed responses are summarized below:
>
> ### On Question 1
>
> We completely share this perspective, which is why we do also have the ROC-AUC scores in Tables 1, 5-8.
>
> Please note that Enrichment Factor (EF) and ROC-AUC score are not alternatives to each other, but rather they are complementary evaluation metrics. Enrichment Factor shows the model performance in finding top-ranking hits, whereas ROC-AUC score compares sensitivity/specificity pair for the ability to identify a drug target. Besides, EF is the standard evaluation metric for benchmarking computational pipelines in virtual screening tasks specific for that domain [Truchon et al 2007]. For example, in all the references in our main tables (Table 1 and 2) as well as the papers mentioned in related work, EF is used as the main performance metric to measure the success of the VS method proposed. This performance metric (i.e. EF) demonstrates the potential of drug candidates in early rankings of the method. This is also known as precision at k metric in statistics.
>
> While many papers published in this domain don’t include ROC-AUC scores, we did report ROC-AUC scores across all drug targets, input modalities and classification models in Tables 1, 5-8. Specifically, Table 1 shows the ROC-AUC scores of ToDD-RF, ToDD-ViT and benchmark models on the Cleves-Jain dataset. Furthermore, Tables 5-6 show the ROC-AUC scores across different domain properties used for graph filtration on the Cleves-Jain dataset using ToDD-RF and ToDD-ViT. Similarly, Tables 7-8 show the ROC-AUC scores across different domain properties used for graph filtration on the DUD-E Diverse dataset using ToDD-RF and ToDD-ConvNeXt. Our ROC-AUC scores outperform the benchmark models and prove the validity of our experiments by combining the measures of sensitivity and specificity, yet the enrichment factor is the key metric in this task. Provided a dataset of thousands or millions of compounds, we are interested to see the top-ranking ligands that can inhibit a drug target, i.e., recognizing active ligands with the highest effectiveness has more practical utility than classifying ligands as active or decoy.
>
> ### On Question 2
> Thank you very much for sharing these references. We added them to our paper (Section 2.1).  First, in our main results (Table 1 and 2), we shared all the state of the art results published in these datasets. Some of them  [7, 36, 42, 83, 91] are using the fingerprinting method. We also compared our method with the most popular fingerprint in the LBVS domain, namely Morgan fingerprints. We have a comprehensive section for this comparison in the Appendix. In Appendix C.5, we give the comparison results with our methods when used with Morgan fingerprints instead of TODD fingerprints. The other fingerprints you mentioned are also used, but Morgan fingerprints are known to outperform them in most datasets. As you mentioned, fingerprinting is a well-studied subject in virtual screening, and in the paper [Capecchi et al 2020], the authors provide an expansive comparison of the performances of these fingerprints with each other. We hope this addresses your question. While we have been constrained primarily by the stringent page limits, we can run and add further experiments with other fingerprints if requested.
>
> ### Citations
> [Capecchi et al 2020] Capecchi, A., Probst, D., & Reymond, J. L. (2020). One molecular fingerprint to rule them all: drugs, biomolecules, and the metabolome. Journal of Cheminformatics, 12(1), 1-15.
>
> [Truchon et al 2007] Truchon, J.F. and Bayly, C.I., 2007. Evaluating virtual screening methods: good and bad metrics for the “early recognition” problem. Journal of chemical information and modeling, 47(2), pp.488-508.

---

> > ### Author Response · Authors · 2022-08-01
> > **Authors' Response to Reviewer WV81 on Question 3**
> >
> > ### On Question 3
> > We thank the referee for this thoughtful point. Imbalanced datasets present inherent complex characteristics and pose several challenges in model training. The most prevalent approaches to address this problem are undersampling the majority class (decoy compounds) and oversampling the minority class (active compounds) by generating synthetic samples using SMOTE techniques. Dealing with class imbalance while training a Random Forest (RF) classifier was actually a more challenging problem in our experiments. We use the general principle of the bootstrap method by resampling the active compounds with replacement, i.e. this is also referred to as overbagging. In addition to that, we tend to regularize the RF classifier by using shallow, pruned trees. On the other hand, our Experimental Results in Section 6.2 show that Triplet Network where Vision Transformer (ViT) serves as the backbone is more robust than the RF classifier on a small scale dataset, i.e. Cleves-Jain, and ensures less variance and less bias across the cross-validation sets.
> >
> > We speculate that there are 2 reasons that cause this phenomenon:
> > 1. First, in comparison to training from scratch, transfer learning by fine-tuning ViT pretrained on ILSVRC-2012 ImageNet improves learning on a small scale, imbalanced dataset by including a larger and balanced auxiliary dataset. In the presence of data imbalance and inadequate number of training samples, transfer learning augments the training data and induces balance into the skewed class distribution [Al-Stouhi et al 2016].
> > 2. Second, utilizing triplet margin loss enforces the network to effectively learn inter-class margins when data in the compound library exhibit highly-skewed class distribution. Unlike cross-entropy loss, weighted cross-entropy loss or focal loss, triplet margin loss mitigates the data imbalance issue by imposing a tighter constraint to preserve the similarity between the compounds from the same cluster in the embedding space.
> >
> > ### Citations
> >
> > [Al-Stouhi et al 2016] Al-Stouhi, Samir, and Chandan K. Reddy. "Transfer learning for class imbalance problems with inadequate data." Knowledge and information systems 48.1 (2016): 201-228.

---

> > > ### Author Response · Authors · 2022-08-01
> > > **Authors' Response to Reviewer WV81 on Questions 4-5**
> > >
> > > ### On Question 4
> > > Thank you very much for pointing out this concern. As you noted above, this is a highly interdisciplinary project (including domain experts from mathematics/topology, applied statistics, engineering, physics, drug discovery & development), which involves deep technical concepts from computational topology, machine learning, and biochemistry. Given the stringent page limits, it was hard to choose which parts to keep in main text and which parts to move into the appendix. We tried our best to make the paper accessible to a wider audience.
> > >
> > > As you suggested, we added more figures to explain the process of obtaining our topological fingerprints.  We added Figure 3 and 4 for sublevel filtration, and Figure 5 for VR-filtration to visually explain the process in the appendix. VR-filtration plays an important role in our construction, but also is the most technical part. With sublevel/superlevel filtrations (vertical direction), we obtain substructures of the compounds induced by chemical functions, e.g., atom weight, ionic radius, partial charge. With VR-filtration (horizontal direction), we capture the information about distances between atoms and sizes of the rings evolving in these substructures, and therefore it provides critical information about the topological characteristics of these substructures in contrast to other methods, which do not provide such granular information [Lim et al 2020, Adams et al 2021]. As can be seen in the toy example in Section B.5, while sublevel/superlevel filtration extracts the induced subgraphs, VR-filtration gives larger graphs by adding edges for each threshold step to capture intrinsic distance information in the data.
> > >
> > > As you recommended, we added a more thorough and clearer explanation why we use VR-filtration (Remark in Appendix Section B.5) and how VR-filtration works with Figure 5. Thank you again for your valuable input and comment.
> > >
> > > ### On Question 5
> > > Thank you for this question. The dimension/size details are given in the description of each method in the appendix Section B.4. In general, the dimension/size depends on how many filtering functions (atom weight, partial charge, bond strength, chirality, etc.) are used, number of thresholds used for these function in the persistence diagrams, and the vectorization method (Betti, Silhouette, Landscape, Persistence Images, etc.) used.
> > >
> > > In particular, for most vectorizations, with 1 filtering function, the output is a matrix of size $m \times r$, where $m$ is the number of thresholds for the filtering function, and $r$ is max threshold for VR-filtration, usually chosen smaller than the largest diameter (distance of farthest two atoms) of the compounds in the dataset. For example, for the Cleves-Jain dataset we chose $r=20$, while DUDE-Diverse, it was chosen as $r=30$. For atom weight function, $m$ is chosen as 10 (10 different thresholds to identify different atom weights in the compounds). This implies for each compound in the Cleves-Jain dataset we get ToDD fingerprints (with atom weight function) as matrices of size 10x20 while for the DUDE-Diverse dataset, we get matrices of size 10x30. In these matrices, 1 direction represents atom weight and the other direction represents the graph distance (VR-filtration). Similarly, if we use 2 filtering functions, say atom weight and bond strength, we would have a 3D array (1 direction atom weight, 1 direction bond strength and 1 direction graph distance (VR) of size 4x10x20 for Cleves-Jain dataset, and 4x10x30 for DUDE-Diverse dataset. Here, 4 comes from the fact that there are 4 different bond types (single, double, triple and aromatic).
> > >
> > > ### Citations
> > > [Lim et al 2020] Lim, S., Memoli, F. and Okutan, O.B., 2020. Vietoris-Rips persistent homology, injective metric spaces, and the filling radius. arXiv preprint arXiv:2001.07588.
> > >
> > > [Adams et al 2021] Adams, H. and Coskunuzer, B., 2021. Geometric Approaches on Persistent Homology. arXiv preprint arXiv:2103.06408.

---

> > > > ### Author Response · Authors · 2022-08-01
> > > > **Authors' Response to Reviewer WV81 on Question 6 and Limitations**
> > > >
> > > > ### On Question 6 & Limitation 1 (Merged)
> > > > We discuss in detail the computational complexity of our model in Section 6.3. Our model is versatile and can be scaled for large libraries by customizing the allocated computational resources. Please note that the analysis in Section C.4 shows the execution time of our computation pipeline when the feature extraction task is distributed across 8 cores of a single Intel Core i7 CPU. It is possible to parallelize computationally costlier operations such as VR-filtration by allocating more CPU cores on the HPC platform and optimize array operations (e.g., numpy) via the joblib library. Furthermore, all ToDD models require substantially fewer computational resources during training compared to current graph-based models that encode a compound through mining common molecular fragments, a.k.a., motifs [Jin et al 2020]. For instance, training a motif based GNN on GuacaMol dataset which has approximately 1.5M drug-like molecules takes 130 hours of GPU time [Maziarz et al 2021]. In contrast, once we generate the topological fingerprints via Vietoris-Rips filtration, training time of ToDD-ViT and ToDD-ConvNeXt for each individual drug target takes less than 1 hour on a single GPU (NVIDIA RTX 2080 Ti).
> > > >
> > > > ### On Limitation 2
> > > > Being a highly interdisciplinary project at the nexus of algebraic topology, ML, and biochemistry, with a team of domain experts (including Mathematics/topology, applied statistics, engineering, physics, drug discovery & development) we show that mathematical foundations behind the model are solid. Since this is a drastically new approach to the Virtual Screening problem, we run extensive experiments in standard benchmark datasets, and we significantly outperform SOTA in all of them with different ML models. This proves our topological fingerprints are model agnostic, and captures highly important information about the substructures induced by the filtering functions used. In the paper, we also compare our performance in both (VS) domain performance metrics, and performance metrics in ML field. We would be happy to provide additional experiments or theoretical justification to enhance the justification rigor.
> > > >
> > > > Thank you very much again for your suggestions and remarks to improve our paper. Please do not hesitate to post further comments or questions.
> > > >
> > > > ### Citations
> > > > [Jin et al 2020] Jin, Wengong, Regina Barzilay, and Tommi Jaakkola. "Hierarchical generation of molecular graphs using structural motifs." International conference on machine learning. PMLR, 2020.
> > > >
> > > > [Maziarz et al 2021] Maziarz, Krzysztof, et al. "Learning to extend molecular scaffolds with structural motifs." arXiv preprint arXiv:2103.03864 (2021).

---

> > > > > ### Comment · Reviewer_WV81 · 2022-08-08
> > > > > **Thank you for the answers.**
> > > > >
> > > > > I'm afraid that the advantage of the proposed method over the existing methods is not clear.

---

> > > > > > ### Author Response · Authors · 2022-08-08
> > > > > > **We demonstrate the advantages of our method against several SOTA Virtual Screening methods as well as the most successful compound fingerprinting technique in Virtual Screening, namely Morgan fingerprints (ECFP4)**
> > > > > >
> > > > > > We demonstrate the robustness of our method over the existing methods with a plethora of empirical evaluations:
> > > > > >
> > > > > > 1. We compare ToDD against 11 Virtual Screening methods across 22 drug targets on the Cleves-Jain dataset, and against 12 Virtual Screening methods across 8 drug targets on the DUD-E Diverse dataset. We outperform state-of-the-art methods on benchmark datasets by a wide and highly statistically significant margin (93% gain for Cleves-Jain and 54% gain for DUD-E Diverse dataset).
> > > > > >
> > > > > > 2. We quantitatively analyze the explainability of our models’ success by replacing topological fingerprints computed via multiparameter persistence with the most popular fingerprinting method: Morgan fingerprints. Morgan fingerprint, also known as extended-connectivity fingerprint ECFP4, is the best performing fingerprint in small molecule virtual screening and target prediction benchmarks.
> > > > > >
> > > > > > **Comparison of EF 2%, 5%, 10% and ROC-AUC values between ToDD and other virtual screening methods on the Cleves-Jain dataset:**
> > > > > > | Model  | EF 2% (max. 50) | EF 5% (max. 20) | EF 10% (max. 10) | ROC-AUC |
> > > > > > | ------------- | ------------- | ------------- | ------------- | ------------- |
> > > > > > | USR | 10.0 | 6.2 | 4.1 | 0.76 |
> > > > > > | GZD | 13.4 | 8.0 | 5.3 | 0.81 |
> > > > > > | PS | 10.7 | 6.6 | 4.9 | 0.78 |
> > > > > > | ROCS | 20.1 | 10.7 | 6.2 | 0.83 |
> > > > > > | USR + GZD | 13.7 | 7.7 | 4.7 | 0.81 |
> > > > > > | USR + PS | 13.1 | 7.9 | 5.0 | 0.80 |
> > > > > > | USR + ROCS | 17.1 | 9.1 | 5.4 | 0.83 |
> > > > > > | GZD + PS | 16.0 | 9.1 | 5.9 | 0.82 |
> > > > > > | PH_VS | 18.6 | NA | NA | NA |
> > > > > > | GZD + ROCS | 20.3 | 10.8 | 5.3 | 0.83 |
> > > > > > | PS + ROCS | 20.5 | 10.7 | 6.4 | 0.83 |
> > > > > > | ToDD-RF | 35.2 ± 2.3 | 15.6 ± 1.0 | 8.1 ± 0.4 | 0.94 ± 0.02 |
> > > > > > | ToDD-ViT | 39.6 ± 1.4 | 18.6 ± 0.4 | 9.9 ± 0.1 | 0.90 ± 0.01 |
> > > > > > | Relative gains | 92.9% | 83.7% | 54.1% | 13.3% |
> > > > > >
> > > > > >
> > > > > > **EF 2% values on Cleves-Jain Dataset using ViT model trained with Morgan fingerprints vs. ToDD fingerprints:**
> > > > > > | Drug Target  | Morgan Fingerprints (ECFP4) | ToDD Fingerprints |
> > > > > > | ------------- | ------------- | ------------- |
> > > > > > | a | 25.0 | 50.0 |
> > > > > > | b | 11.4 | 34.1 |
> > > > > > | c | 3.8 | 46.2 |
> > > > > > | d | 50.0 | 50.0 |
> > > > > > | e | 10.0 | 30.0 |
> > > > > > | f | 37.5 | 50.0 |
> > > > > > | g | 30.0 | 50.0 |
> > > > > > | h | 40.0 | 50.0 |
> > > > > > | i | 20.0 | 50.0 |
> > > > > > | j | 17.9 | 21.4 |
> > > > > > | k | 14.3 | 39.3 |
> > > > > > | l | 45.0 | 35.0 |
> > > > > > | m | 38.9 | 50.0 |
> > > > > > | n | 15.0 | 35.0 |
> > > > > > | o | 13.3 | 20.0 |
> > > > > > | p | 2.2 | 32.6 |
> > > > > > | q | 18.2 | 18.2 |
> > > > > > | r | 14.3 | 32.1 |
> > > > > > | s | 10.0 | 33.3 |
> > > > > > | t | 20.0 | 50.0 |
> > > > > > | u | 27.8 | 50.0 |
> > > > > > | v | 35.7 | 42.9 |
> > > > > > | Mean | 22.7 | 39.5 |
> > > > > > | ROC-AUC | 0.86 | 0.90 |
> > > > > >
> > > > > > **Comparison of EF 1% (max. 100) between ToDD and other virtual screening methods on 8 targets of the DUD-E Diverse subset:**
> > > > > > | Model | AMPC | CXCR4 | KIF11 | CP3A4 | GCR | AKT1 | HIVRT | HIVPR | Avg. |
> > > > > > | ------------- | ------------- | ------------- | ------------- | ------------- | ------------- | ------------- | ------------- | ------------- | ------------- |
> > > > > > | Findsite | 0.0 | 0.0 | 0.9 | 21.7 | 34.2 | 39.0 | 1.2 | 34.7 | 16.5 |
> > > > > > | Fragsite | 4.2 | 42.5 | 0.0 | 32.9 | 29.1 | 47.1 | 2.4 | 48.7 | 25.9 |
> > > > > > | Gnina | 2.1 | 15.0 | 38.0 | 1.2 | 39.0 | 4.1 | 11.0 | 28.0 | 17.3 |
> > > > > > | GOLD-EATL | 25.8 | 20.0 | 33.5 | 17.9 | 34.6 | 29.2 | 28.7 | 23.4 | 26.6 |
> > > > > > | Glide-EATL | 35.5 | 20.8 | 30.5 | 15.1 | 24.0 | 31.6 | 29.0 | 22.0 | 26.1 |
> > > > > > | CompM | 32.3 | 25.0 | 35.5 | 33.6 | 37.1 | 44.2 | 30.2 | 25.0 | 32.9 |
> > > > > > | CompScore | 39.6 | 51.6 | 51.3 | 14.0 | 27.1 | 37.6 | 21.8 | 18.2 | 32.7 |
> > > > > > | CNN | 2.1 | 5.0 | 11.2 | 28.7 | 12.8 | 84.6 | 12.2 | 9.9 | 20.8 |
> > > > > > | DenseFS | 14.6 | 5.0 | 4.3 | 44.3 | 20.9 | 89.4 | 12.8 | 8.4 | 25.0 |
> > > > > > | SIEVE-Score | 30.7 | 61.1 | 53.4 | 6.7 | 33.3 | 42.1 | 39.8 | 38.3 | 38.2 |
> > > > > > | DeepScore | 28.1 | 56.8 | 54.3 | 37.1 | 40.9 | 59.0 | 43.8 | 62.8 | 47.9 |
> > > > > > | RF-Score-VSv3 | 32.3 | 60.9 | 4.5 | 25.9 | 32.5 | 41.9 | 39.8 | 65.7 | 37.9 |
> > > > > > | ToDD-RF | 42.9 ± 4.5 | 92.3 ± 3.2 | 75.0 ± 5.0 | 67.6 ± 3.4 | 78.9 ± 4.0 | 90.7 ± 1.3 | 64.1 ± 2.3 | 92.1 ± 1.5 | 73.7 |
> > > > > > | ToDD-ConvNeXt | 46.2 ± 3.6 | 84.6 ± 2.8 | 72.5 ± 3.6 | 28.8 ± 2.8 | 46.0 ± 2.0 | 81.2 ± 2.5 | 37.5 ± 3.6 | 74.6 ± 1.0 | 58.9 |
> > > > > > | Relative gains | 16.7% | 51.1% | 38.1% | 52.6% | 92.9% | 1.5% | 46.3% | 40.2% | 53.9% |
> > > > > >
> > > > > > **EF 1% values and ROC-AUC scores on DUD-E Diverse dataset using ConvNeXt model trained with Morgan fingerprints (ECFP4) vs. ToDD fingerprints:**
> > > > > > | Model | EF 1% (Morgan) | ROC-AUC (Morgan) | EF 1% (ToDD) | ROC-AUC (ToDD) |
> > > > > > | ------------- | ------------- | ------------- | ------------- | ------------- |
> > > > > > | AMPC | 38.5 | 0.87 | 46.2 | 0.81 |
> > > > > > | CXCR4 | 48.0 | 0.97 | 84.0 | 0.99 |
> > > > > > | KIF11 | 57.5 | 0.95 | 72.5 | 0.97 |
> > > > > > | CP3A4 | 20.5 | 0.84 | 28.8 | 0.91 |
> > > > > > | GCR | 46.7 | 0.94 | 46.0 | 0.97 |
> > > > > > | AKT1 | 60.0 | 0.98 | 81.2 | 0.98 |
> > > > > > | HIVRT | 50.0 | 0.96 | 37.5 | 0.95 |
> > > > > > | HIVPR | 61.3 | 0.98 | 74.6 | 0.99 |
> > > > > > | Mean | 47.8 | 0.94 | 58.8 | 0.95 |

---

> > > > > > > ### Author Response · Authors · 2022-08-08
> > > > > > > **We demonstrate the advantages of our method against several SOTA Virtual Screening methods as well as the most successful compound fingerprinting technique in Virtual Screening, namely Morgan fingerprints (ECFP4)**
> > > > > > >
> > > > > > > We hope that our response is sufficiently informative for you to reconsider assessing a higher rating for our paper.
> > > > > > >
> > > > > > > We have demonstrated that our topological compound descriptors are model-agnostic and have proven to be exceedingly competitive, yielding state-of-the-art results unequivocally over all baselines.
> > > > > > >
> > > > > > > Our extensive experimental evaluations show comparisons between ToDD and several SOTA VS methods. Moreover, we present an ablation study replacing topological fingerprints computed via multiparameter persistence with the most successful compound fingerprinting method in VS, namely Morgan fingerprints [Capecchi et al 2020].
> > > > > > >
> > > > > > > ### Citations
> > > > > > > [Capecchi et al 2020] Capecchi, Alice, Daniel Probst, and Jean-Louis Reymond. "One molecular fingerprint to rule them all: drugs, biomolecules, and the metabolome." Journal of cheminformatics 12.1 (2020): 1-15.

---

> > > > > > > > ### Author Response · Authors · 2022-08-09
> > > > > > > > **Thank you for your consideration**
> > > > > > > >
> > > > > > > > We provided detailed responses, pointing to the specific parts of the paper that articulates the points you expressed concerns. We would very much appreciate if you could consider updating the scores before the deadline today.

---

### Official Review · Reviewer_sEym · 2022-07-12

**Rating:** 7
**Confidence:** 3
**Soundness:** 3 good
**Presentation:** 3 good
**Contribution:** 4 excellent

**Summary:**

In this paper, a new type of molecular fingerprint is introduced, and its application in the virtual screening setup is shown. The fingerprint is based on multiparameter persistence homology that produces topological descriptions of molecules in the form of multidimensional vectors. The methodology expands single persistence methods and applies multiple filtering functions, which can be based either on atom distances or chemical properties. Architectures such as Vision Transformer and ConvNeXt are used to process the multidimensional fingerprints. Finally, the triplet loss is used to train these models for the virtual screening task. The method is compared against many baselines on two datasets. Additionally, the computational complexity is calculated, and the stability of the multiparameter persistence fingerprints is formally proven in the supplementary material.

**Questions:**

1. How were the atomic properties (atomic mass, partial charges) selected for the evaluation? Could you visualize graph hierarchies w.r.t. these filtration functions for some example compounds?
2. In section 6.1 it is said that “ViT is more robust to distinct arrangements of atoms in space, also referred to as molecular conformation.” Are conformations encoded in the fingerprint? Based on the description in section 4, the fingerprints use only graph edge distances.
3. How is the triplet loss used to predict the ligand ranking, which is used to calculate EF scores?


**Limitations:**

In Appendix D, only positive societal impacts are presented. The computational complexity is a limitation that was analyzed in Section 6.3, but not mentioned in the section about method limitations. The number of atomic properties used as filtering functions not only increases the cost of calculating the fingerprint, but also requires more memory and processing power of the ML models.

**Strengths And Weaknesses:**

Strengths:
- The theoretical foundations of the paper are sound. The persistence homology is thoroughly explained, and the modifications are presented using a clear mathematical notation (though some definitions may be found only in the supplementary materials).
- The stability of the proposed fingerprint is formally proven.
- The experimental evaluation on two datasets shows the strong performance of the proposed method. The number of baseline models in the comparison is impressive.
- A section about computational complexity was included in the paper.
- The ablation study corroborates the usefulness of multiparameter persistence and explains architectural choices.

Weaknesses:
- Unfortunately, the method is not presented in a clear way. Some of the details were moved to the supplementary materials, so it is difficult to grasp the idea before reading the appendix and rereading the paper. For example, VR-filtration is not explained in the main text, but this term is used multiple times. Maybe it would be a good idea to add a figure explaining how the fingerprint is calculated (showing VR-filtering on one axis and property-based filtering on the other one).
- The proposed method shows a great improvement over the other method included in the comparison, but these methods are mostly scoring functions that were trained with a significantly different objective. Please, correct me if I am wrong, but I think that most of these methods use docking poses to score ligands and this is how they build their rankings. The MP Fingerprint, on the other hand, is trained to separate active ligands from decoys only based on the topological features and ignoring the 3D conformations. It would be useful to add a few simple baselines, e.g. a random forest model on Morgan fingerprints (similar to the ablation study, but I cannot find the information about which model was used in this experiment).
- Section 6.4 mentions a comparison of more contrastive objectives, but the (quantitative) results of the preliminary experiments are not included in the paper.
- The code is not provided, which makes the results difficult to reproduce.

Minor comments:
- In the abstract it is said that “VS is used for comparing a library of compounds against known active ligands to identify drug candidates…” This description of VS is inaccurate as it is not the only way to perform virtual screening. For example, there are structure-based approaches to the VS that can be performed even if no active ligands are known.
- It would be interesting to see the performance of the MP fingerprints in different benchmarks, e.g. ADMET property prediction.

---

> ### Author Response · Authors · 2022-08-01
> **Authors' Response to Reviewer sEym on Weakness 1**
>
> We thank the reviewer for the time and effort spent in reviewing our paper, and for the detailed suggestions. Reflecting your valuable comments, we made a major revision adding multiple figures in Appendix Section B.5 to facilitate the understanding of graph filtration and computation of multiparameter (MP) persistence using Vietoris-Rips (VR) simplicial complexes. Additionally, we shared the code that is relevant to reproduce results. We believe that our paper was significantly improved, thanks to your comments. We hope that our responses would be sufficiently informative for you to reconsider assessing a higher rating for the revised paper.
>
> Detailed responses are summarized below:
>
> ### On Weakness 1
>
> Thank you very much for pointing out this concern. As you noted above, this is a highly interdisciplinary project (with domain experts from mathematics/topology, applied statistics, engineering, physics, drug discovery & development), which involves deep technical concepts from computational topology, machine learning, and biochemistry. Given the page limits, it is challenging to choose which parts to keep in main text and which parts to move into the appendix. We tried our best to make the paper accessible to a wider audience, especially relevant to this conference. Per your suggestion, we added more figures to explain the process of obtaining our topological fingerprints.  In particular, in a toy example (Figure 4) we explain how filtering with two chemical functions (i.e., atom weight and partial charge) induce the relevant substructures of a small compound, cytosine (Figure 3). In another toy example (Figure 5), we explain how Vietoris-Rips complexes capture topological information of these substructures by using graph distances. We revised Table 10 and 11 to simplify the comparison. We further made several minor edits to clarify exposition. If you have further suggestions for reorganizing the paper, we are happy to make revisions as requested within the constraints of the 9-page limit.
>
> We added Figure 3 and 4 for sublevel filtration, and Figure 5 for VR-filtration to visually explain the process in the appendix. VR-filtration plays an important role in our construction, but also is the most technical part. With sublevel/superlevel filtrations (vertical direction), we obtain substructures of the compounds induced by chemical functions, e.g., atom weight, ionic radius, partial charge. With VR-filtration (horizontal direction), we capture the information about distances between atoms and sizes of the rings evolving in these substructures, and therefore it provides critical information about the topological characteristics of these substructures in contrast to other methods, which do not provide such granular information [Lim et al 2020, Adams et al 2021]. As can be seen in the toy example in Section B.5, while sublevel/superlevel filtration extracts the induced subgraphs, VR-filtration gives larger graphs by adding edges for each threshold step to capture intrinsic distance information in the data. As you recommended, we added a more thorough and clearer explanation why we use VR-filtration (Remark in Appendix Section B.5) and how VR-filtration works with Figure 5. Thank you again for your valuable input and comment.
>
> ### Citations
> [Lim et al 2020] Lim, S., Memoli, F. and Okutan, O.B., 2020. Vietoris-Rips persistent homology, injective metric spaces, and the filling radius. arXiv preprint arXiv:2001.07588.
>
> [Adams et al 2021] Adams, H. and Coskunuzer, B., 2021. Geometric Approaches on Persistent Homology. arXiv preprint arXiv:2103.06408.

---

> > ### Author Response · Authors · 2022-08-01
> > **Authors' Response to Reviewer sEym on Weaknesses 2-4**
> >
> > ### On Weakness 2
> >
> > Thank you for this thoughtful question. In [Capecchi et al 2020], the most common fingerprints in VS are compared.  Yes, some of the baselines we cited are Structure-Based Virtual Screening (SBVS) Methods [7, 91, 42, 36, 83], and they “also” use the target information as well as the active ligand information. However, as mentioned in the experiments section (Section 6.1), we use exactly the same settings (5-fold cross-validation) as other methods to ensure fair comparison. All LBVS and SBVS methods cited in our main tables (Table 1 and 2) use the same number of ligands in their training set. However, by our Ligand-Based Virtual Screening (LBVS) approach, we focus on producing an effective fingerprint of the compounds where the ML tools can work very efficiently. In other words, our approach makes ML tools work more effectively by converting the classification task into a suitable form to study.
> >
> > In general, this question can also be formulated as a choice between LBVS vs. SBVS. In theory, some suggest SBVS is better than LBVS in general, while others believe the opposite is true [please see Ripphausen et al 2010, Yang et al 2019 for discussion]. In general, SBVS methods are computationally more costly than LBVS methods. Therefore, SBVS could be very effective on specific targets with small compound libraries. But for large libraries with more general tasks, LBVS appears to be more effective with the recent advances in machine learning. Gains in performance yielded by our approach also supports this conclusion.
> >
> > In our ablation study (Appendix Section C.5), by using the Vision Transformer (ViT) and ConvNeXt models, we compared the performance of Morgan Fingerprints and our TODD Fingerprints on both Cleves-Jain dataset (Table 10) and DUD-E Diverse dataset (Table 11). The results show that our fingerprints outperform the most popular fingerprint in the field for these benchmark datasets.
> >
> > ### On Weakness 3
> >
> > In Section 6.4, we tested a number of ablations of our model to analyze the effect of its individual components and to further investigate the effectiveness of our topological fingerprints. One of them is investigating the network architectures and loss objectives to determine the most suitable one for deep metric learning. Our code has a flexible implementation with several switch cases and offers a user-friendly command-line interface to test the robustness of different network architectures and loss objectives such that a Siamese network trained with a contrastive loss or a Triplet network trained with circle loss can be easily compared against our model choice: a Triplet network trained with triplet margin loss. We run several experiments to choose the most suitable deep metric learning technique; however quantifying the results of different network architecture choices for all the drug targets from both the Cleves-Jain and DUD-E Diverse datasets using 5-fold cross-validation requires training 135 different models.
> >
> > ### On Weakness 4
> >
> > We strive for reproducibility but this work is a result of an interdisciplinary collaboration between academia (Mathematics/Topology, Applied Statistics, Engineering, Physics) and industry (Drug Discovery & Development). While we outline the details of the method in the paper, the source code could not be shared because of legal and IP related constraints. We agree that the reproduction of the work would be easier with an accompanying code. Following the referee’s suggestion, we are sharing a version of the code to ensure that our results are trustworthy and reproducible.
> >
> > ### Citations
> > [Ripphausen et al 2010] Ripphausen, P., Nisius, B., Peltason, L. and Bajorath, J., 2010. Quo vadis, virtual screening? A comprehensive survey of prospective applications. Journal of medicinal chemistry, 53(24), pp.8461-8467.
> >
> > [Yang et al 2019] Yang, X., Wang, Y., Byrne, R., Schneider, G. and Yang, S., 2019. Concepts of artificial intelligence for computer-assisted drug discovery. Chemical reviews, 119(18), pp.10520-10594.
> >
> > [Capecchi et al 2020] Capecchi, A., Probst, D., & Reymond, J. L. (2020). One molecular fingerprint to rule them all: drugs, biomolecules, and the metabolome. Journal of Cheminformatics, 12(1), 1-15.

---

> > > ### Author Response · Authors · 2022-08-01
> > > **Authors' Response to Reviewer sEym on Questions 1-3**
> > >
> > > ### On Question 1
> > > Thank you very much for this suggestion. In Figure 4, we visualize how filtering with two chemical functions (atom weight and partial charge) induce the relevant substructures of a small compound, cytosine (Figure 3). In our experiments, we mainly used atomic weight, partial charge, and bond strength as filtering functions as they are independent of each other, and together they give a very fine filtering of the datasets. Note that in addition to these filtering functions, we always use VR-filtration (graph distance) in one of the directions in our ToDD fingerprints. Since this problem is ligand similarity search for general dataset, we used these most common functions. However, depending on the target protein, these functions can be customized for more targeted information extraction. In particular, for molecular prediction problems, these functions should be chosen very carefully so that the induced substructures indeed relate to the molecular property studied. There are several functions to be used for this goal like ionic radius, chirality, aromaticity, orbital hybridization, number of bonded Hydrogen atoms, etc. In a follow-up project, we aim to expand our efforts in this particular direction.
> > >
> > > ### On Question 2
> > > We do not generate molecular conformations. Technically, persistence diagrams resulting from Vietoris-Rips filtrations use the distance matrix, which contains the geodesic distances taken pairwise between the atoms of a compound. CNNs are known to have inductive bias and they heavily rely on the existence of a certain type of spatial structure in the data. In comparison to CNNs, ViT has much less spatial inductive bias, since it cuts the input sample into patches and encodes positional embeddings that allow the model to retain positional information and encode positionally invariant relationships [Dosovitskiy et al 2020]. This concept can be particularly useful in the task of property prediction where distinct molecular conformers impact the property to estimate, e.g., prediction of atomization energies relies heavily on electrostatic interaction between the nuclei in 3D space.
> > >
> > > ### On Question 3
> > > For training, given the multiparameter Vietoris-Rips persistent homology matrices of 3 compounds (anchor, positive and negative), Vision Transformer (ViT) or ConvNeXt models provide embedding vectors of the anchor sample (reference input), positive sample (ligand that inhibits the same drug target with anchor) and negative sample (ligand that is inactive against a drug target). Triplet margin loss enforces the order of distances by minimizing the Euclidian distance from the anchor to the positive input and maximizing the Euclidian distance from the anchor to the negative input. Its formulation is similar to hinge loss requiring a soft margin with a slack variable
> > > $\alpha$:
> > > $L(x, x^{+}, x^{-})=\max(0, \alpha + \lVert \mathbf{f(x)-f(x^{+})} \rVert_{2} - \lVert \mathbf{f(x)-f(x^{-})} \rVert_{2})$.
> > >
> > > This objective has been demonstrated to offer enhancements of embedding in learning to rank tasks [Wang et al 2019]. Hence we use the triplet margin loss to finetune the parameters of the pretrained networks.
> > >
> > > In order to predict the ligand ranking:
> > >
> > > 1. We compute the multiparameter Vietoris-Rips persistent homology features of the test compounds and feed them to the trained ViT or ConvNeXt model to acquire their embeddings.
> > > 2. The embedding of each ligand from the compound library is compared against the embedding of the anchor sample to predict the ligand ranking. Specifically, we measure the similarity score using the inverse of the Euclidean distance between the embedding of an anchor and drug candidate. This similarity score is applied to predict the ligand ranking and calculate EF scores.
> > >
> > > We thoroughly elaborated this concept in Section 6.1 in the Enrichment Factor paragraph.
> > >
> > > ### Citations
> > > [Dosovitskiy et al 2020] Dosovitskiy, Alexey, et al. "An image is worth 16x16 words: Transformers for image recognition at scale." arXiv preprint arXiv:2010.11929 (2020).
> > >
> > > [Wang et al 2019] Wang, Xinshao, et al. "Ranked list loss for deep metric learning." Proceedings of the IEEE/CVF conference on computer vision and pattern recognition. 2019.

---

> > > > ### Author Response · Authors · 2022-08-01
> > > > **Authors' Response to Reviewer sEym on Limitations and Minor Comments**
> > > >
> > > > ### On Computational Limitations
> > > > Thank you very much for pointing this out. Yes, you are right, increasing the number of filtering functions is definitely adding to the computational cost. When used for large libraries, this can degrade the execution performance. We added this limitation (given below) to Appendix D.2.
> > > >
> > > > We discuss in detail the computational complexity of our model in Section 6.3. Our model is versatile and can be scaled for large libraries by customizing the allocated computational resources. Please note that the analysis in Section C.4 shows the execution time of our computation pipeline when the feature extraction task is distributed across 8 cores of a single Intel Core i7 CPU. It is possible to parallelize computationally costlier operations such as VR-filtration by allocating more CPU cores on the HPC platform and optimize array operations (e.g., numpy) via the joblib library. Furthermore, all ToDD models require substantially fewer computational resources during training compared to current graph-based models that encode a compound through mining common molecular fragments, a.k.a., motifs [Jin et al 2020]. For instance, training a motif based GNN on GuacaMol dataset which has approximately 1.5M drug-like molecules takes 130 hours of GPU time [Maziarz et al 2021]. In contrast, once we generate the topological fingerprints via Vietoris-Rips filtration, training time of ToDD-ViT and ToDD-ConvNeXt for each individual drug target takes less than 1 hour on a single GPU (NVIDIA RTX 2080 Ti).
> > > >
> > > > ### On Minor Comment 1
> > > > Thank you very much for pointing out this oversight. We corrected the phrase as follows:
> > > >
> > > > In computer-aided drug discovery (CADD), virtual screening (VS) is used for identifying the drug candidates that are most likely to bind to a molecular target in a large library of compounds.
> > > >
> > > > ### On Minor Comment 2
> > > > Thank you very much for this valuable suggestion. Indeed, our follow-up and more extensive project is to study the problem of “molecular property prediction” by adapting our model to regression/classification tasks.
> > > >
> > > > Thank you very much again for your suggestions and remarks to improve our paper. Please do not hesitate to post further comments or questions.
> > > >
> > > > ### Citations
> > > > [Jin et al 2020] Jin, Wengong, Regina Barzilay, and Tommi Jaakkola. "Hierarchical generation of molecular graphs using structural motifs." International conference on machine learning. PMLR, 2020.
> > > >
> > > > [Maziarz et al 2021] Maziarz, Krzysztof, et al. "Learning to extend molecular scaffolds with structural motifs." arXiv preprint arXiv:2103.03864 (2021).

---

> > > > > ### Comment · Reviewer_sEym · 2022-08-08
> > > > > **Thank you!**
> > > > >
> > > > > I appreciate your detailed responses and addressing my comments in the revised manuscript. I find the current text more clear and approachable. I also appreciate including raw data in the python research notebooks although the method implementation remains confidential.
> > > > >
> > > > > I have one more question regarding Figure 4, which is a great addition to the paper. Should not both filtration criterions be considered jointly (as a logical conjunction)? In the second row from the top, an oxygen atom appears in columns 3 and 4 while its atomic number is 8 (should be lower or equal 7). Similarly, hydrogen atoms attached to carbons have partial charges greater than 0.1 according to Figure 3, but they appear in the second column at the top.

---

> > > > > > ### Author Response · Authors · 2022-08-08
> > > > > > **Thank you very much!**
> > > > > >
> > > > > > We thank the reviewer for the time and effort spent in reviewing our paper, and for the detailed suggestions. We greatly appreciate that you noticed the typo in Figure 4. We fixed it and submitted a revision.
> > > > > > We believe that our paper was significantly improved thanks to your comments. We hope you would find the usefulness of our methodology on topological ML to accelerate drug discovery, and consider higher rating.

---

> > > > > > > ### Comment · Reviewer_sEym · 2022-08-08
> > > > > > > **Rating re-evaluation**
> > > > > > >
> > > > > > > I decided to raise the rating because all my concerns were accurately addressed, especially the clarity of presentation. Congratulations for your solid research work!

---

> > > > > ### Author Response · Authors · 2022-08-08
> > > > > **Thanks a lot!**
> > > > >
> > > > > Thanks so much! We are very grateful for your constructive and detailed response, your appreciaticion of our work and, of course, for raising the score!

---

### Meta-Review · Area_Chair_nGKK · 2022-08-23

**Recommendation:** Accept
**Confidence:** Certain

**Metareview:**

The reviewers mostly liked the paper. They mentioned the sound theoretical foundations, stability, and strong empirical performance. The rebuttal was able to convince most reviewers that the paper should be accepted for NeurIPS.

**Award:**

No

---

### Decision · Program_Chairs · 2022-09-14

Accept